# “Neuroimmunoendocrinology” in Children with Rheumatic Diseases: How Glucocorticoids Are the Orchestra Director

**DOI:** 10.3390/ijms241713192

**Published:** 2023-08-25

**Authors:** Maria Cristina Maggio, Angela Miniaci, Romina Gallizzi, Adele Civino

**Affiliations:** 1University Department PROMISE “G. D’Alessandro”, University of Palermo, Via del Vespro 129, 90100 Palermo, Italy; mariacristina.maggio3@gmail.com; 2Paediatric Rheumatology, UOC of Paediatrics, IRCCS Azienda Ospedaliero-Universitaria di Bologna, 40138 Bologna, Italy; 3Paediatric Unit, Department of Health Science, Magna Graecia University of Catanzaro, 88100 Catanzaro, Italy; rgallizzi@unicz.it; 4Paediatric Rheumatology and Immunology, Vito Fazzi Hospital, 73100 Lecce, Italy; adelecivino@gmail.com

**Keywords:** glucocorticoids, rheumatic diseases, growth hormone, puberty, IGF-1, growth plate, bone, teeth, cytokines

## Abstract

The neural, the endocrine, and the immune systems are studied as distinct districts in physiological and pathological settings. However, these systems must be investigated with an integrative approach, while also considering that therapeutic agents, such as glucocorticoids, can induce a reversible or irreversible change of this homeostasis. Children and adolescents affected by rheumatic diseases frequently need treatment with corticosteroids, and the treatment must sometimes be continued for a long time. In the biological era, the treat-to-target strategy allowed a real revolution in treatment, with significant steroid dose sparing or, in many patients, steroid treatment withdrawal. In this review, the impact of glucocorticoids on endocrine, immune, and neurologic targets is analyzed, and the crosstalk between these systems is highlighted. In this narrative review, we explore the reasoning as to why glucocorticoids can disrupt this homeostasis, we summarize some of the key results supporting the impact of glucocorticoids treatment on endocrine, immune, and neurologic systems, and we discuss the data reported in the international literature.

## 1. Introduction

Historically, researchers have defined the neural, the endocrine, and the immune systems as distinct districts to be studied for physiology and diseases. However, these systems must be investigated with an integrative approach, since the homeostasis of the whole body depends on the interactive roles played by each of them and, in synergy, by all of them.

In this narrative review, we explore the reasoning as to why glucocorticoids can disrupt this homeostasis and induce a reversible or irreversible change in the integrity of each system, we summarize some of the key results supporting the impact of glucocorticoids treatment on endocrine, immune, and neurologic systems, and we discuss the data reported in the international literature.

To acquire more data on the topic, we performed a literature review to identify articles of potential interest about glucocorticoids effects on immune, endocrine, and neurologic systems in children and adolescents with rheumatic diseases. We searched PubMed to find relevant articles published in English, with no restrictions regarding the publication year, up to 1 March 2023. The following search strings were used: ‘glucocorticoid’ [Title/Abstract] AND ‘growth hormone’ [Title/Abstract] OR ‘IGF-1′ [Title/Abstract] OR ‘growth plate’ [Title/Abstract] OR ‘bone’ OR ‘teeth’ [Title/Abstract] OR ‘puberty’ [Title/Abstract] OR ‘cytokines’ [Title/Abstract] OR ‘rheumatic diseases’ [Title/Abstract] AND ‘pediatric’ [Title/Abstract]. We also reviewed references from relevant articles on this topic, not identified in the original search.

Although large clinical cohort studies evaluating the impact of glucocorticoids on the close interaction of endocrine, immune, and neurologic systems in children with rheumatic diseases are lacking, there are several reports suggesting that steroids have these effects, and that a therapeutic dose-sparing approach is the goal of treat-to-target and personalized therapy [1].

## 2. A Strict Dialog among Neurotransmitters, Hormones, and Cytokines

The relationships linking the nervous, endocrine, and immune systems are outstanding, and several shared mechanisms of action, mediators, and receptors are considered main actors of these links. These factors contribute to create a network linking the different systems in a plastic and still only partially known way. In fact, hormones and cytokines are both key mediators of the hypothalamus–pituitary–adrenal (HPA), hypothalamus–pituitary–IGF-1, and hypothalamus–pituitary–thyroid (HPT) axes as a reaction to inflammation and stress, driving immune modulation. The activation of the HPA axis increases the synthesis and the release of hypothalamic corticotropin-releasing hormone (CRH), increasing the release of the pituitary adrenocorticotropic hormone (ACTH) and the stimulation of cortisol and dehydroepiandrosterone (DHEA) secretion by the adrenal glands (Figure 1). An imbalance of this hormonal pathway affects both stimulating and inhibiting regulatory loops, causing an increased risk and/or worsening of many infectious diseases [2].

An additional relevant neuro-endocrine path, a milestone to deal with stress, is the sympathoadrenal system. It is a principal pathway of communication between the central nervous system and the immune system. The role of stress in the pathogenesis of autoimmune diseases has been attributed to glucocorticoids, end-point mediators of HPA, and to catecholamines, end-point messengers of sympathoadrenal system, comprising the sympathoneural axis, of which the mediator is noradrenaline, and the sympathoadrenal axis, of which the mediator is adrenaline [3]. Furthermore, cells regulating innate immunity and T lymphocytes can secrete noradrenaline that may be involved in a local autocrine and paracrine self-amplifying feed-forward loop enhancing proinflammatory cytokines synthesis by myeloid cells and inflammatory damage in experimental autoimmune encephalomyelitis (EAE) and in multiple sclerosis (MS). All cell types involved in the autoimmune inflammation and targeted tissue damage in autoimmune central nervous system diseases express receptors for catecholamines [4]. Catecholamines exert most of their immunomodulatory actions through β_2_-adrenoreceptors; however, α-adrenoreceptors can play a role in the diseases [4].

Noradrenaline and adrenaline stimulate the β_2_-adrenoreceptor–cAMP–protein kinase A pathway and inhibit type 1 proinflammatory cytokines secretion, such as interferon-γ, tumor necrosis factor (TNF)-α, and interleukin (IL-12) by antigen-presenting cells and T helper (Th) 1 cells. However, they stimulate the synthesis of type 2 anti-inflammatory cytokines (IL-10). Through this mechanism, systemically, endogenous catecholamines may selectively inhibit Th1 and cellular immunity, and may shift Th2 toward dominance of humoral immunity [5]. Nonetheless, noradrenaline and adrenaline can promote IL-8, IL-1, and TNF-α secretion in specific conditions, where inflammation acts as a local reaction to triggers and focus the response by neutrophils chemotaxis and other specific pathways [2].

Both psychological and physical stressors are relevant triggers of autoimmune diseases such as MS. The altered secretion of adrenaline and noradrenaline is documented to precede (in patients affected by MS) and/or arise during the development of MS and EAE, as a relay to immune cell activation (in the early phase of the disease) and an injury of sympathoadrenal system neurons and of their projections (in the late phase of the disease) [4].

The 24-h urinary adrenaline levels were in the normal range in patients with systemic lupus erythematosus (SLE) or with rheumatoid arthritis (RA). Prednisolone at therapeutic doses of 30–40 mg/day significantly suppressed adrenaline secretion. These data suggest that the human adrenal medulla secretes adrenaline dependently on the HPA axis [6].

An enhanced activity of the sympathetic nervous system (SNS) was described in adult patients with SLE and with RA. In these patients, decreased levels of ACTH and cortisol were documented, as well as in patients not yet treated with glucocorticoids. These data suggest a reduced tone of the HPA axis in patients with SLE or RA. Furthermore, the low levels of cortisol in relation to hyperactivity of CNS neurotransmitters may play a proinflammatory role, since a joining of the two endogenous response axes is lacking [7].

The different organs and systems must interact to maintain homeostasis, and the analogies among immune, nervous, and endocrine systems are outstanding. These systems show a crosstalk involving many shared mediators, hormones, cytokines, and receptors in physiological and in pathological settings. In fact, hormones and cytokines are main actors of the HPA and HPT axes, with feedback to stress and promotion of immune modulation.

## 3. Corticosteroids, Exogenous Glucocorticoids, and Rheumatic Diseases in Children and Adolescents

Glucocorticoids remain a milestone of rheumatic disease treatment. Children and adolescents affected by rheumatic diseases frequently need treatment with corticosteroids, and the treatment must sometimes be continued for a long time.

However, DMARDs allow significant dose sparing of corticosteroids. Furthermore, in the biological era, the treat-to-target strategy allows the real revolution in treatment, with significant steroid dose sparing or, in many patients, steroid treatment withdrawal.

Recent advances in the management of rheumatic and autoinflammatory diseases have permitted obtaining disease remission in many patients. Hence, future treatment guidelines must include the target to reach clinical remission or, in some cases, minimal disease activity. The improvement of treatment approaches allows targeting and maintaining the strict disease control, for instance, with a bridging treatment with a short cycle of low-dose oral glucocorticoids (i.e., prednisone at a dosage of 5–10 mg/day), after starting DMARDs or biologic agents. The adoption of the treat-to-target therapeutic strategy may improve disease outcome and reduce the glucocorticoid cumulative dose, using the minimal dose for the briefest time, until the goal is reached [8,9].

Monthly monitoring in active disease is necessary to check and reduce the short-term glucocorticoid course, waiting for the therapeutic effect of DMARDs, to verify whether children were tolerating the drugs and to evaluate side effects.

In recent times, pediatric rheumatologists have developed descriptions of disease states that correspond to appropriate therapeutic targets, such as inactive disease, minimal disease activity, and parent- or child-acceptable symptom states. The goal of treatment remains tailored therapy, considering, case by case, the benefits or risks of glucocorticoids. A tailored therapy may be improved by a treat-to-target approach, employing glucocorticoids for the minimal needed time, recognizing the early window of opportunity to start biologic drugs, adapting the treatment to the patients’ priority needs, providing support, and encouraging adherence [10].

Pediatric rheumatologists’ interest in glucocorticoids treatment is linked to their impact on metabolic homeostasis, nutrition, hormones secretion, immune system response, and organ and tissue development. In fact, both synthetic and endogenous steroids diffuse across the cell membrane, selectively bind glucocorticoids receptors, and modulate their functions.

For their significant and systemic impact on bone, growth, puberty, protein synthesis, and lipid and glucose metabolism, corticosteroid side effects are fearsome in the pediatric population [8].

The feedback of glucocorticoid secretion is mediated by the HPA axis with a circadian rhythmicity. Corticosteroid secretion is through the adrenal cortex from cholesterol, with three pathways of secretion: cortisol (a glucocorticoid hormone, produced by the zona fasciculata, regulates fat, protein, and carbohydrate metabolic pathways, inhibits inflammation, regulates blood pressure, and decreases bone mineralization) and aldosterone (a mineralocorticoid hormone, produced by the zona glomerulosa) [11]. Aldosterone regulates sodium and potassium blood levels, blood pH, stimulating the excretion of magnesium, potassium and hydrogen ions and the reabsorption of sodium ions by the distal tubules and the nephron collecting duct. Aldosterone further regulates blood pressure via the renin–angiotensin–aldosterone axis. Aldosterone sends signals to the kidneys. Adrenals also secrete, via the zona reticularis, sexual androgens (DHEA, delta-4- androstenedione, etc.), with a mild activity. These hormones contribute to the secondary sexual characteristics during puberty [12] (Figure 1).

Further organs (such as the thymus, brain, skin, and intestine) express the enzymes required for corticosteroid synthesis and can synthetize biologically active corticosteroids. The local tissue and organ synthesis of corticosteroids exhibits paracrine and autocrine feedback and regulates homeostasis, immune cell activation, and differentiation. The thymus, for instance, synthesizes corticosteroids in epithelial cells and developing thymocytes, contributing a maintenance of adequate immunological activity [13].

The glucocorticoid receptor (GR) mediates the biological effects of glucocorticoids. The NR3C1 gene encodes the GR protein, part of the nuclear receptor superfamily of ligand-dependent transcription factors [14]. The NR3C1 gene encodes the GRα and the GRβ, two isoforms with high homology, obtained by the alternative splicing of the last exon. In fact, GR contains nine exons, and the last exon represents the difference between the two GR isoforms; GRα includes exon 9α (50 amino acids), and GRβ incorporates exon 9β (15 amino acids). The GR isoforms additionally expand the range of glucocorticoid responses in physiological and pathological conditions.

GRα expression differs in many tissues, and it is significantly higher than GRβ expression in most cell types [15]. GRα exists in the cell’s cytoplasm, in the absence of the ligand; after corticosteroid binding, GR is activated and is carried into the nucleus. It operates as a transcription factor. In fact, GR binds to the DNA, controlling the transcription of several genes via direct or indirect DNA binding. The direct binding is realized by GR homodimers binding to GR response elements (GREs), allowing gene transcription regulation by the GR. On the other hand, GRβ is expressed in the nucleus, does not bind corticosteroids, is inactive as a transcription factor, and hampers GRα transcriptional activity.

GR inhibition of many genes, involved in the proinflammatory response, happens through synthesis induction of anti-inflammatory proteins, as well as through inhibition of proinflammatory transcription factors, e.g., activator protein-1 (AP-1) and nuclear factor-kappa B (NF-kappaB), via a mechanism defined as “trans repression”. The metabolic and undesired effects of GR are usually exerted via a “transactivation” mechanism [8,14].

Furthermore, a wide number of functionally different GR subtypes are subject to several post-translational changes that further control the cellular response to glucocorticoids.

Children exposed to high doses of glucocorticoid may develop resistance to the drug, with a loss of immune suppression. The resistance is secondary to many pathogenetic mechanisms, such as GR downregulation, GR polymorphisms, defective GR translocation, and increased expression of GRβ, inhibiting GRα [8,15,16].

Cortisol, the main endogenous glucocorticoid in humans, circulates in the blood, predominantly bound to corticosteroid-binding globulin (CBG). CBG supports cortisol circulation and regulates the release to tissues [17]. CBG-free cortisol can passively go across the plasma membrane, and its cellular bioavailability is regulated by two enzymes: 11β-hydroxysteroid dehydrogenase type 1 (11β-HSD1) converts cortisone to cortisol; 11β-hydroxysteroid dehydrogenase type 2 (11β-HSD2) oxidizes cortisol into cortisone, the inactive metabolic form. In fact, the activity of these two enzymes influences the cellular variabilities in glucocorticoid sensitivity [18,19].

However, most drugs containing glucocorticoids are not bound to CBG and are not metabolized by 11β-HSD2. These biological characteristics distinguish therapeutic glucocorticoids from endogenous cortisol and contribute to explaining the impact that glucocorticoids have on growth, puberty, and bone mineralization, including in children who receive low doses of glucocorticoids, but for a long time [19].

In fact, synthetic glucocorticoids, particularly dexamethasone, show higher affinity and greater bioavailability than endogenous hormones (cortisol), which bind poorly to corticosteroid-binding globulin. They have a plasma half-life much longer than cortisol [20] (Table 1).

The main difference is the typical circadian and ultradian (with a pulsatile profile) rhythm of endogenous hormones, which allows their biological effects, promoting GR activation and reactivation as a response to the ultradian endocrine secretion, driven by fast reusage of GR linkage to chromatin-binding sites as a relay to the hormonal spikes, coupling the promoter activity to the physiologic fluctuations of hormone concentrations [12].

Moreover, the signaling pathway of GR shows a rapid and well-timed response to hormone level variations, demonstrating that the regulation precision of gene targets by GR needs an ultradian pattern of hormone stimulation [15] (Figure 2).

The pulsatile profile is not guaranteed by exogenous glucocorticoids, which fail to guarantee an ultradian activity of GR on chromatin and accordingly fail to couple hormone pulses with the transcriptional response. Therefore, the transcriptional regulation can be deeply altered by synthetic glucocorticoids or even by natural hormones when they are not administered respecting the natural pattern [22].

Cytokines are strong stimulators of the HPA axis and can permanently change the endogenous glucocorticoid secretion when they are increased in early life [23]. HPA axis activation increases not only plasma cortisol levels, but also plasma cortisone levels. Cortisone shows a weak binding to GR, but is available in plasma (with insignificant binding to CBG) and can be rapidly converted inside the cells to active cortisol by 11β-HSD1.

## 4. Growth Delay and Short Stature

Children with juvenile idiopathic arthritis (JIA) manifest a short stature with a variable incidence, depending on the subtype of JIA; short stature or growth and puberty delay is described in 10.4% of children with polyarticular JIA and in 41% of children with systemic JIA (sJIA). Children with oligoarticular JIA develop increased growth of bones of the involved limbs, with premature growth plate fusion. These patients remodel shorter limbs, when they reach definitive stature [24].

Overall, 41% of children with persistent oligoarticular JIA manifest a mild growth restriction, which is more severe in children who receive several intra-articular glucocorticoid injections. Earlier use of DMARDs in children requiring repeated intra-articular glucocorticoid injections [25] may reduce growth restriction.

During glucocorticoid therapy, children with sJIA experience a mean loss of height in SDS of −2.7 +/− 1.5, with a direct correlation with glucocorticoid therapy extent. After glucocorticoid discontinuation, slow linear growth persists in 30% of children. Their adult height is associated with the severity of growth delay during the disease’s active phase and with height catch-up growth after the remission [26].

In these cases, early treatment with DMARDs, intra-articular glucocorticoids, and/or biologics contributes to prevent limb dysmetria [18].

The pathogenesis of growth delay is multifactorial: chronic inflammation, undernutrition, long-term treatment with glucocorticoids, altered body composition, hypogonadism, or puberty delay. These elements influence the GH/IGF-1 axis and the GnRH–gonadotropin–gonadic axis. Furthermore, they have an influence on the growth plate cellular proliferation and maturation [27].

The new therapeutic frontiers can better control inflammation; however, 10–20% of children with severe forms of JIA still maintain persistent growth deficiency, with a final short height [24].

A cross-sectional study, involving girls with JIA, showed that patients with polyarticular JIA and higher cumulative glucocorticoid doses more likely had a short stature. This study suggests that, even six months after glucocorticoid treatment interruption, children with polyarticular or systemic JIA are still inclined to short stature and delayed puberty [28].

A cohort of children with JIA, treated before or during the biologic era, showed a significant better long-term disease activity and damage when their disease onset occurred in the biologic era. Growth failure was reported in patients with oligoarticular and in a higher number of patients with polyarticular JIA with the onset before the biologic era. Pubertal delay was reported only in patients with polyarticular JIA with the disease onset before the biologic era [29].

Chronic inflammation causes decreased height velocity and growth failure to variable degrees, depending on hypersecretion of proinflammatory cytokines, such as IL-1, IL-6, and TNF-α.

IL-6 inhibits chondrocyte proliferation, and chondrocytes can synthetize IL-6 in response to inflammatory and physiologic stimuli.

Cytokines are also synthetized in the growth plate, where chondrocytes and synoviocytes are a source of IL-6. Hormonal factors and cytokines such as IGF-1 and 2, testosterone, estradiol, insulin, platelet growth factor, IL-1, TNF-alpha, and IFN-gamma increase IL-6 secretion. On the contrary, hydrocortisone significantly reduces IL-1-induced IL-6 synthesis [30].

However, other elements are crucial cofactors, such as the disease severity, disease progression, malnutrition, kidney and/or liver insufficiency, glucocorticoid treatment, and endogenous hypercorticism. All these factors act via inhibition of the GH–IGF-1 axis and IGFBP-3 synthesis and decrease growth plate development.

The height at 3 years after initial clinical presentation was within the SDS for sex and age in a cohort of children with JIA. However, children with JIA showed a decrease in growth rate in the first 3 years after disease onset. Overall, 39% of children showed a growth deficiency early at disease onset, more severe in patients with sJIA (24%) or psoriatic JIA (PsA) (26%). A significant growth failure in children who have not been treated with corticosteroids is documented [31], proposing that growth restriction in sJIA is multifactorial and may be strictly correlated to the systemic inflammatory condition. In fact, high levels of IL-6 in children with sJIA may significantly decrease growth velocity.

Late diagnosis and treatment are correlated with lower SDS for height at presentation. However, children with the lowest SDS for height at the diagnosis had the highest increase in stature at 3 years [32].

Children with a longer disease history, a longer treatment time with systemic glucocorticoids, and a lower BMI showed a lower SDS for stature. A noticeable predictor of short stature at diagnosis was the functional disability at first presentation, evaluated by validated tests [33] and compared with clinometry. JADAS for children with JIA and sJADAS in children with sJIA demonstrate good measurement assets and are valid instruments for the valuation of disease activity [34]. In further studies, JADAS and sJADAS can be included in the auxological follow-up of patients with JIA, to predict their final height.

An associated factor, which contributes to growth delay, is delayed puberty and/or hypogonadism, shown by many patients with a low control of the disease and persistent inflammation. These patients have an adjunctive cause of peak bone mass reduction, with possible concomitant or future bone fragility [24].

### 4.1. Glucocorticoids and Growth

Long-term glucocorticoid treatment causes a short stature via different mechanisms: inhibition of hypothalamic GHRH secretion [35], inhibition of adenohypophysis secretion of GH, and peripheral IGF-1 and IGFBP-3 secretion.

The exposure to an excess of glucocorticoids inhibits GH secretion with growth delay in childhood and abnormalities in bone mineral density (BMD) and body composition in both children and adults.

A significant linear growth reduction is described in about 10–20% of patients with JIA, mainly in those suffering from systemic or polyarticular forms of the disease and requiring high doses of glucocorticoids, and for periods longer than 1 year [36]. Glucocorticoid therapy slows growth when the dosage is at least 0.25 mg/kg/day prednisone-equivalent [26].

However, cortisol physiologically contributes to the maturation and function of somatotrophs, supporting the hypothesis that glucocorticoids act with a dose-dependent effect on the somatotropic axis.

Studies in vivo confirm the role of glucocorticoids as stimulators and inhibitors of GH secretion, and the real biologic result depends on the hormonal concentrations and on the time of exposure. Furthermore, glucocorticoids have a positive role in GH gene activation and expression [37].

The strict interaction between GH and glucocorticoids is highlighted by the influence of GH and IGF-1 on the biological activity of glucocorticoids regulating 11β-hydroxysteroid dehydrogenase type 1 activity, thus influencing the peripheral metabolism of cortisol. In fact, the GH–IGF-I axis inhibits the expression and the activity of 11beta-HSD1 in liver and in adipose tissue, resulting in decreased cortisol regeneration [38].

The pattern of GH secretion after glucocorticoids shows a distinctive profile, showing the triphasic short-term effects of exogenous glucocorticoids on the stimulation of GH secretion. Glucocorticoids acutely decrease GH secretion. This action is probably mediated by somatostatin, secreted by the hypothalamus. After that, sustained acute hypercortisolemia, persisting for at least 3 h, achieved with the oral administration of dexamethasone, a long-acting strong synthetic glucocorticoid, stimulates a transient increase in the serum levels of GH. This secretion pattern can be secondary to the reduced negative feedback of IGF-1 and the increased GHRH secretion, in opposition to the acute somatostatin pulse after a short-term exposure to glucocorticoids. The GH response to GHRH is inhibited 12 h after dexamethasone administration [39]. However, patients who receive multiple doses of glucocorticoids have persistent hypercortisolemia, with a possible persistent somatostatin increase, which could inhibit GH secretion. Many children with rheumatological diseases, treated with long-term (>3 months) glucocorticoids (mainly prednisolone and methylprednisolone) experience a reduction in, but not the complete absence of, GH secretion.

The effect of exogenous glucocorticoid pharmacological doses on GH secretion has been investigated in patients with autoimmune or allergic diseases and in recipients of an organ transplantation. In these patients, an excess of glucocorticoids reduced, but did not completely block, the GH response to several stimuli and reduced spontaneous GH secretion. Hence, the primary effect of the chronic glucocorticoid excess seems to increase the somatostatin hypothalamic secretion [40]. These children show a reduced growth rate.

Other than the influence on GH secretion, glucocorticoids have a direct impact on growth plates, inhibiting GH receptor expression, IGF-1 synthesis, chondrocyte mitosis, collagen synthesis, and cartilage sulfation. All these mechanisms are possible cofactors of growth impairment.

These pathogenetic factors are the possible rationale for GH treatment in selected adolescents with a severe growth impairment and a documented GH deficiency, who present a poor final height prognosis [41]. However, these patients may need high GH doses, as observed in children with a genetic origin of growth impairment, as SHOX haploinsufficiency [42].

### 4.2. Glucocorticoids and the Bone

Glucocorticoids inhibit calcium bowel absorption and kidney reabsorption, contributing to osteopenia and osteoporosis. Moreover, secondary pediatric osteoporosis is more frequent than expected in children and adolescents affected by rheumatic diseases and can be secondary to many causes such as chronic malnutrition and/or malabsorption, eating disorders, endocrine diseases, neuromuscular disabilities, nephropathies, liver diseases, chronic inflammation, and glucocorticoid treatment.

Children with chronic cholestatic diseases, secondary to autoimmune diseases associated with sclerosing cholangitis, for instance, show a decreased bone mass gain; an impaired hepatic IGF-1 synthesis can be the pathogenetic link to their low bone mass.

Chronic liver disease (both non-cholestatic and cholestatic) can induce malabsorption of vitamin D and calcium, and the lacking vitamin D 25-hydroxylation, with secondary osteopenia and osteoporosis. Vitamin D deficiency triggers secondary hyperparathyroidism (hepatic osteodystrophy). Likewise, chronic kidney disease, secondary to SLE or to other autoimmune diseases, causes “renal osteodystrophy”, part of the “chronic kidney disease—mineral bone disorder” (CKD-MBD) [43]. Children with CKD-MBD show a reduced BMD, decelerated bone maturation, short stature, and bone deformities, with bowed legs, secondary to the decrease in phosphorus and calcium serum levels. CKD-MBD is documented in most of patients receiving dialysis treatment, and it has been documented in children with kidney disease before dialysis begins. Kidney or liver transplantation is not followed by a satisfactory BMD gain, because the use of glucocorticoids and immuno-suppressants does not support the full recovery.

Children with prolonged immobilization, such as limited mobility secondary to JIA, experience an impaired physiological mechanical stress on bone, with inhibited osteoblast-mediated bone formation and accelerated osteoclast-mediated bone resorption, consequently inducing a reduction in BMD. In addition, a low dietary intake of vitamin D and calcium is a further risk factor for pediatric osteoporosis [44].

Glucocorticoids disrupt trabecular bone architecture, and their primary actions are on osteoblasts and osteocytes. In fact, glucocorticoids inhibit osteoblasts replication, differentiation, and activity, induce mature osteoblast and osteocyte apoptosis, and activate osteoclast genesis, suppressing bone formation and strongly decreasing BMD and bone quality [45].

Hence, iatrogenic Cushing’s syndrome, secondary to glucocorticoid treatment, induces structural and functional derangement of bone metabolism, secondary to a decreased osteoblast number and function, as supported by decreased levels of alkaline phosphatase and osteocalcin.

These patients have a high incidence of fractures (30–50%), frequently involving the spine. Patients with a recent diagnosis show an annual incidence of vertebral fracture of 4–6%; the prevalence in patients with a long disease course many years after the diagnosis is 7–28%. Most of the fractures are asymptomatic, involving the thoracic district. The fracture risk is significantly higher in children affected by systemic diseases associated with severe inflammation such as dermatomyositis, SLE, and sJIA. Neither glucocorticoid dose nor BMD is an ideal risk predictor of fractures, which recognizes a multifactorial pathogenesis, including muscle involvement. Children and adolescents with rheumatic diseases have an increased risk of long-bone fractures, principally of the wrist and of the forearm. In these children, long-bone fractures are not predictive of vertebral fractures. Physiological BMD increase, as in healthy peers, is reduced across the years, although the use of DMARDS and biological drugs can reverse the trend caused by disease activity and glucocorticoid treatment [46,47].

High-dose protocols with methotrexate (MTX) can decrease BMD in children with malignancies; nonetheless, the low doses of methotrexate, prescribed as DMARDs in JIA, are not correlated with a BMD loss. In contrast, the therapeutic effect on inflammation in vivo balances the inhibitory effects on osteoblasts in vitro [48].

Moreover, vitamin D insufficiency may contribute to the disease and require therapeutic integration in children with rheumatic diseases [49,50].

Children and adolescents with SLE have a high risk of developing osteoporosis later in life, because they get the disease before the peak of BMD is achieved, with a low BMD for the rest of life.

Osteopenia, measured by dual-energy X-ray absorptiometry (DEXA), is reported in 40% of these patients, and the site more severely affected is the lumbar spine. Children with SLE do not experience a bone mass catch-up in youth. Glucocorticoids are correlated with a reduced bone mass in these patients; thus, pediatric rheumatologists choose to prescribe corticosteroid doses down to the lowest possible dose and for the shorter period, whenever possible [51]. The new biologic therapies, which permit a steroid-sparing treatment, will also have advantageous effects on bone health in children and adolescents with SLE [52].

In other conditions, such as dermatomyositis, a major role in low BMD is played by reduced mobilization [53].

Osteoporotic fractures are a relevant cause of morbidity in children with rheumatic diseases, treated with glucocorticoids. High disease activity, body mass index z-scores, average daily glucocorticoid doses, and lumbar spine BMD z-scores, worsening in the first 6 months, predicted the risk of accidental vertebral fractures over 6 years. Overall, lumbar spine BMD z-scores persisted low at 6 years, consistent with inadequate recovery of bone health [54].

There is no proven safe association between the glucocorticoid dose and the risk of fractures; in fact, a risk of osteoporosis is also described with 2.5 mg/day. Daily mean dose is associated with an increased risk of fractures; cumulative glucocorticoid doses are associated with decreased BMD. Patients treated with higher doses (≥7.5 mg/day of prednisolone or equivalent) show an increased risk of non-vertebral fracture, hip fracture, and vertebral fracture relative to patients treated with lower doses of glucocorticoids (<2.5 mg/day).

These data suggest that the adverse effects of glucocorticoids are rapidly manifested and are correlated to daily dose. The cumulative dose of glucocorticoid exposure does not strongly influence the risk of fracture [55]. An increase of 0.5 mg/kg in the daily glucocorticoid dose is associated with a 2.1-fold amplified risk of vertebral and non-vertebral fractures over 6 years [54].

Preventive strategies against corticosteroid-induced osteoporosis are necessary to prevent low BMD and fractures, soon after starting the glucocorticoid therapy [55].

Furthermore, many factors (gender, age, pubertal stage, baseline BMD, underlying disease, comorbidities, and poorly controlled underlying disease) contribute to determining glucocorticoid-induced osteoporosis [56].

A study on 833 patients aged >18 years, who fulfilled the 2010 ACR/European League Against Rheumatism classification criteria for rheumatoid arthritis (RA) diagnosis, treated with glucocorticoids during the study period, compared BMD to the control group, including patients diagnosed with RA not treated with glucocorticoids for >1 year. No significant differences in the changes in BMD and incidental fractures between the two groups was reported. The disease activity score for 28 joints with erythrocyte sedimentation rate was the only parameter inversely correlated to BMD changes. Thus, the benefits of glucocorticoids reducing inflammation compensate for the risk of osteoporosis in patients with RA, if appropriate strategies to prevent bone loss are promoted [57].

Another study on patients with RA treated with glucocorticoids, evidenced that a dosage of prednisone ≤ 5 mg/day was not associated with a BMD decrease. This result is probably due to the control of inflammation by glucocorticoids, which mitigates the effect on the risk of osteoporosis. However, doses > 7.5 mg/day were negatively correlated with BMD only in patients with moderate to high disease activity [58].

We can speculate that, in the pediatric population, glucocorticoids can have more influence, because the BMD peak has yet to be reached.

In addition, a long-term treatment with glucocorticoids, especially with high doses, induces reduced growth velocity, as well as pubertal and bone age delay. These factors amplify the risk of fractures.

Glucocorticoid-induced osteoporosis is a reversible disease, secondary to several factors. However, it shows a slow recovery, requiring 10 or more years [49], thus exposing these children and adolescents to a high risk of fractures with persistently low lumbar spine BMD z-scores [49,50].

However, it is still a matter of study how and how much rheumatic diseases in childhood and adolescence can induce secondary osteoporosis by themselves. Patients with rheumatic diseases and chronic inflammation also show low BMD before the use of glucocorticoids. Further bone loss and osteoporosis risk factors are associated with the primary disease, the severity of inflammation, and comorbidities [51,54,58]. In fact, it is reported that children with rheumatic diseases acquired vertebral fractures in the first year of glucocorticoid therapy and a significant reduction in BMD, being more severe in children with severe systemic inflammation. Furthermore, rheumatic diseases, particularly polyarticular JIA and sJIA, are associated with vertebral fractures, also before glucocorticoid treatment [56,59].

These data are in part supported by the finding that JIA patients have a lower cortical BMD than controls, that SLE and JIA patients show a reduced trabecular BMD, and JIA patients have a decrease in muscle area compared to SLE [60]. In these patients, the promotion of physical activity improved BMD and could recover their body posture, as described in children with chronic diseases, affecting bone maturation. In fact, body posture is a feedback–feedforward loop in which the CNS acquires afferent signals from the postural receptors, processes and integrates these inputs, and transmits an efferent signal to the postural tonic system. Changes in the musculoskeletal system alter postural balance and allow osteoporosis or decreased muscle tone. Otherwise, a physical fitness program can counteract the weight gain often associated with glucocorticoid treatment and a low-physical-activity lifestyle [61,62].

Proinflammatory cytokines, such as TNF-alpha, IL-1, and IL-6, cause inflammation-associated osteoporosis by affecting the differentiation and functional role of osteoblasts and osteoclasts, and uncoupling the bone remodeling cycle, with a secondary negative bone balance [63]. Furthermore, cytokines stimulate osteoclastogenesis directly, by operating on cells of the osteoclast lineage, and indirectly, by modulating the synthesis of key molecules, such as RANKL, in the target cells. Cytokines upregulate RANK in the osteoclast precursors and improve their sensitivity to RANKL concentrations [64]. In fact, IL-1 beta and IL-6 promote osteoblasts to synthesize RANKL, promoting bone marrow stem-cell differentiation into osteoclasts and, thus, increasing bone resorption [65].

Nonetheless, B and T lymphocytes, mainly T helper 17 (Th-17), contribute to regulate bone remodeling. Th-17 cells secrete a specific cytokine pattern with a prevalence of IL-17, a strong osteoclast activator, while the secretion of IFN-gamma and IL-4 (inhibitors of osteoclastogenesis) is low [66].

Clinical control of disease activity by the association of etanercept with methotrexate, in children with polyarticular JIA nonresponders to methotrexate, allows a prompt catch-up growth, a reduction in disease activity, and an improvement of body composition and BMD [67].

Another effect of glucocorticoids on bone is the risk that systemic high-dosage corticosteroids trigger osteonecrosis (ON) and result in osteochondral (OC) lesions. Furthermore, the effect of intra-articular corticosteroid injections (IACIs) on joint cartilage and subchondral bone can be associated with OC.

The pathophysiology of glucocorticoid-induced osteonecrosis is still a matter of study. Many factors contribute to this complication: angiogenesis inhibition, hypercoagulability, fat cell hypertrophy, osteoblast and osteocyte apoptosis, increased intraosseous pressure, and decreased blood flow, secondary to bone edema and fat cell hypertrophy. Children with rheumatic diseases such as SLE and JIA have a relatively high incidence of OC lesions, in atypical locations and with an early presentation. The severity of disease activity, as well as systemic and local inflammation, might be intrinsic risk factors for OC lesions. However, repetitive steroid injections need to be considered an associated risk factor in these patients [68].

OC lesions are more frequent in knees and hips in patients who received multiple steroid injections and/or high doses of systemic glucocorticoids but are associated with many risk factors typical of the underlying disease, such as inflammation and mechanical stress.

A further factor is the incidence of specific joint involvement in JIA. A study on 95 children with JIA revealed a significant difference in the involvement of specific joints in persistent oligoarticular- and polyarticular-onset JIA patients with arthritis after a 5-year follow-up. Patients showed arthritis of the shoulder [11% vs. 37%; *p* = 0.026], elbow [26% vs. 63%; *p* = 0.003], wrist [22% vs. 51%; *p* = 0.024], metacarpophalangeal joint [4% vs. 56%; *p* < 0.001], proximal interphalangeal joint [4% vs. 47%; *p* < 0.001], hip [7% vs. 33%; *p* = 0.019], and ankle [59% vs. 98%, respectively; *p* < 0.001]. Joint involvement of the knee [89% vs. 98% respectively; *p* = 0.291] was comparable between the persistent oligoarticular- and polyarticular-onset JIA patients [69]. Hip involvement must undergo an accurate differential diagnosis for the risk of a neoplastic origin of the joint disease [70].

Disease-related features, including Raynaud’s phenomenon and vasculitis, increase the risk of ON lesions among patients with SLE [71]. Furthermore, age at the time of the start of glucocorticoids is associated with ON in pediatric patients with SLE [72].

### 4.3. Glucocorticoids and the Teeth

It has been demonstrated that ameloblasts have a specific pathway of hormonal receptors, variable with the patient’s developmental stage. GR is significantly expressed and contributes to amelogenesis regulation, and the highest level of GR is found in maturation-stage ameloblasts. In fact, glucocorticoids involve enamel mineralization and hardness, supporting the hormonal control of late steps of enamel mineralization and, hence, of enamel quality instead of enamel quantity [73]. Glucocorticoid treatment can interfere with amelogenesis, contributing to the development of molar and incisor hypomineralization (MIH). Several retrospective studies have found an association of early-childhood antibiotic, acetaminophen, FANS, and glucocorticoid use with MIH. The GR appears as the common element able to regulate the expression of enamel crucial genes, regulating enamel synthesis or guiding enamel hypomineralization in the case of interference. The involved teeth can reveal signs of post-eruptive enamel breakdown, and patients have discomfort due to the severe hypersensitivity [74]. Thus, children with severe MIH have limitations in achieving everyday activities, such as consummation of hot or cold foods, tooth brushing, talking, and smiling. Hence, these lesions may be misdiagnosed as caries and undergo radiographic analysis. In these cases, as in other infancy teeth diseases, radiographic study is not mandatory for the diagnosis [75].

### 4.4. Iatrogenic Cushing’s Syndrome

Weight gain, iatrogenic Cushing’s syndrome, and suppression of the HPA axis, with possible progression to adrenal insufficiency (AI), are side effects of glucocorticoid treatment and depend on the dose and the treatment period.

Weight gain may additionally be secondary to the increase in appetite and to the positive effect of food such as gastric juice secretion tamponade, in patients treated with glucocorticoids.

Abdominal obesity and moon face are documented in all patients; AI may occur in most of these patients when treatment is stopped too fast [76].

Some patients also experience these effects with treatment longer less than 2 months. Some patients also show these effects with low doses of glucocorticoids; however, they are rare with doses lower than the daily need for glucocorticoids (5.7–7.4 mg/m^2^/day).

### 4.5. Glucocorticoids and Inflammation

The hypothalamus responds to several signals such as circadian fluctuations, psychological and physical stress, and tissue trauma, which activates the HPA axis via the stimulation of CRH and arginine vasopressin (AVP), with the prompt glucocorticoid secretion increase. Oxytocin may increase ACTH release, whereas atrial natriuretic peptide inhibits ACTH release via direct feedback on the pituitary. Glucocorticoids act on the hypothalamic and pituitary receptors to suppress the release of AVP, CRH, and ACTH in response to these neuropeptides [77]. The hypothalamus is likewise stimulated by cytokines (IL-1, IL-6, and TNF-α). Toll-like receptor 2 (TLR2) and TLR4 activation on adrenocortical cells further stimulates steroidogenesis [78].

Glucocorticoids inhibit the HPA axis via negative feedback on the hypothalamus and pituitary gland, and via the inhibition of proinflammatory cytokines [79]. These parallel inhibitory loops highlight the strict correlation between immune system and adrenal axis. Glucocorticoids inhibit the transcription of several genes encoding proinflammatory chemokines, cytokines, enzymes, and cell adhesion molecules, and they are involved in the start and/or persistence of the inflammatory response in the host. This is the key factor underlying their anti-inflammatory and immune suppression activity, as cortisol and cortisone, prednisone, and prednisolone are also substrates for the 11β-HSD enzymes, modulating their access to GR.

Children and adolescents with rheumatic diseases, requiring long-term treatment with glucocorticoids, are at high risk of developing obesity, especially in cases of limited physical activity. These patients show the overexpression of 11β-HSD1 in adipose tissue, related to obesity.

Immune cells express both GR and 11β-HSD1; however, the expression is strongly dependent on cell differentiation and the activation state. 11β-HSD1 is low or absent in circulating leukocytes, negligible in monocytes, and low in lymphocytes and neutrophils. Monocytes, following differentiation to macrophages or dendritic cells, express a significant increase in 11β-HSD1 activity, which is further enhanced by polarization to “M1” proinflammatory macrophages, compared to “M2” anti-inflammatory macrophages [80]. Furthermore, the increase of endogenous glucocorticoids by 11β-HSD1 may affect the macrophage state and IL10, a cytokine with immune modulatory functions secreted by monocytes and B, Th2, and Treg lymphocytes, is a documented GR target gene in macrophages and contributes to the anti-inflammatory effects of glucocorticoids [81,82].

In rheumatic diseases, 11β-HSD1 is greatly expressed at the sites of inflammation, where it converts inactive glucocorticoids to active metabolites [83]. Hence, 11β-HSD1 plays an essential role in the control of synovitis, joint destruction, and systemic BMD loss in JIA.

The inhibition of immune response by glucocorticoids depends on many points. The inhibition of antigen-presenting cell activation and differentiation, the reduction in proinflammatory cytokine secretion, basophil apoptosis, and the enhanced accumulation of neutrophils at the sites of inflammation are among the mechanisms of glucocorticoid immunosuppression of innate immunity. Parallelly, the adaptative immunity is modulated by glucocorticoids, via immature and mature T lymphocyte apoptosis and polarization from a Th1 to Th2 phenotype. Glucocorticoids also inhibit B lymphocytes. Immature B lymphocytes are more responsive to glucocorticoid-induced apoptosis than mature B lymphocytes. In fact, B lymphocytes express the GR throughout their development, and the GR regulates various transcription factors (NFAT, AP-1, and NF-κB) downstream of B cell receptor (BCR) signaling, explaining the impact of glucocorticoids on B lymphocyte selection [8,19].

Variable blood lymphocyte count is inversely correlated with nictemeral fluctuation of glucocorticoid secretion, and there is a 40% reduction in the number of circulating T lymphocytes from night to morning because of modified tissue homing [84].

The effects of long-term glucocorticoid treatment on vaccination efficacy are poorly investigated, and just a modest impact on immunoglobulins secretion has been described. However, glucocorticoids may promote the production of IgE, while the other immunoglobulin isotype concentrations are unchanged or decreased. Glucocorticoid-mediated IgE increase may exert a direct effect on B lymphocytes, whereby glucocorticoids cooperate with IL-4 to promote isotype class switching, in addition to indirect effects on B lymphocytes via the actions of glucocorticoids on T lymphocytes and monocytes. The efficacy of glucocorticoids in the treatment of atopic patients, nevertheless, suggests that glucocorticoid-mediated increase of IgE production does not worsen the disease [85].

Corticosteroids induce upregulation of genes implicated in immunosuppression and inhibition of chemotaxis, such as IL-10, CD1d, decoy receptor for IL-1R, IL-1 receptor antagonist, and FOXP3. Moreover, corticosteroids stimulate the transcription of inhibitors of proinflammatory regulators, such as KLF2 and IκBα. The mechanism of upregulation of genes able to control inflammation is via DNA indirect binding of GR to genes encoding transcription factors, with the inhibition of chemokines, proinflammatory cytokines (TNF-α, IL-1β, IL-2, IL-3, and IL-6), enzymes (cyclooxygenase 2, nitric oxide), and adhesion molecules [16,19].

However, the time of exposure to glucocorticoids and glucocorticoid dose are key factors of their response. In fact, glucocorticoids upregulate the expression of genes active in the recognition of pathogens and tissue trauma signals, such as pattern recognition receptors (PRRs), complement factors, chemokine and cytokine receptors, and scavenger receptors, while they suppress the expression of proinflammatory cytokines, chemokines, and genes involved in adaptive immunity, such as genes encoding TCR components and genes involved in T cell activation, comprising MHC class II genes. The proposed model [79] suggests that glucocorticoids regulate the immune system with a biphasic pathway; low doses promote the expression of genes regulating innate immune system and rapid responses to insults, whereas high concentrations, induced by stress or pharmacological doses, suppress immune receptors signaling. Patients with a sufficient glucocorticoid secretion can yield a prompt immune response to pathogens and tissue injury; nonetheless, this response shows a controlled extent. In glucocorticoid-insufficient patients, however, subnormal expression of cytokine receptors and PRRs contribute to a slower immune response and to the absence of glucocorticoid-mediated inhibition of immune receptor signaling, contributing to the prolonged duration of the immune response [79].

GR target genes include TLR2, TLR4, the inflammasome component NOD-, pyrin domain-containing 3 (NLRP3), which may alert cells to PAMP and DAMP signals. They increase the expression of several cytokine receptors, including the receptors for IFN-γ, TNF-α, IL-1, and IL-6 [16]. The prompt glucocorticoid increase triggered by physiological stress can act as a systemic warning scheme and sensitize cells to inflammatory cytokines, PAMPs, and DAMPs. Therefore, glucocorticoids (in association with catecholamines) are key actors of the impaired immune system associated with chronic stress [86]. In fact, single and combined actions of the most important stress hormones, noradrenaline, adrenaline, and cortisol, mediating specific leukocyte subpopulation mobilization or trafficking, are determinant in the immune response, which directs leukocyte subpopulations to specific target organs during stress, and significantly enhances the rapidity, efficacy, and control of the immune response [87].

After acute stress, an increase in the bloodstream of neutrophils, B lymphocytes, and T lymphocytes is followed by a reduction in cells outside the blood circulation, apart from neutrophils that also increase in damaged tissues. The strict talk between the endocrine and immune systems regulates immune response, which is empowered in tissues enriched with leukocytes and suppressed in districts depleted of leukocytes during and following stress. Stress hormones play a pivotal role in the redistribution of immune cells: noradrenaline and adrenaline mobilize cells into the bloodstream, while adrenaline and cortisol activate cells traffic to lymphoid tissues and tissues with damage or inflammation [88].

These complex interactions also strictly involve CNS. In fact, IL-1 and/or TNF-α inhibition hampers sleep in children, whereas the increase in IL-1 and/or TNF-α, documented in the brain after sleep deprivation, improves NREMS. These data document the role of cytokines in the regulation of brain function and sleep rhythm, by mediators such as GHRH, prostaglandins, and nuclear factor kappa B. On the other hand, anti-inflammatory cytokines such as IL-4, IL-10, and IL-13 counteract this effect [89].

## 5. Discussion

In the last few years, the strict cooperation of the CNS, endocrine, and immune systems have been investigated. However, many points of this crosstalk are still a matter of debate. Further studies can contribute to define the best strategy to mimic the personalized cortisol circadian and ultradian profile of patients who need glucocorticoid treatment, to guarantee less adverse events and the minimum impact on growth, puberty, and BMD.

The knowledge of the genetic background of GR polymorphisms, the molecular properties of GR isoforms, and their role in the sensitivity and specificity of the glucocorticoid response could contribute to further personalize therapeutic choices.

The biologics era for the treatment of JIA, autoinflammatory syndromes, and autoimmune diseases has dramatically changed the prognosis of the affected children, nonetheless increasing concerns on the potential risk of infections and other adverse drug events in these patients [90]. The risk is higher in children with prolonged glucocorticoid treatment, since the combined immune suppressant action of these drugs can increase the incidence of opportunistic infections. These concerns need to be considered by a multispecialist team, able to guarantee a safe and integrate patient care.

The risk is increased in the first months of life, when the infant has not yet completed the vaccination schedule. Further studies must investigate the age-related specific impact and side effects of glucocorticoids in the different pediatric age groups.

Adolescents need special attention, and further efforts are required to tailor glucocorticoid therapy, in a period of life when ensuring growth and puberty is an essential health need. Catch-up growth can be obtained if glucocorticoids are stopped before final stature is reached: a goal of the tailored therapy is to guarantee the target height, avoiding the risk of short stature, an element with a heavy impact on quality of life. Furthermore, when the BMD peak is not achieved during puberty, it is lost, and cannot be recovered later in their life, with a higher risk of fractures. The “lesson” of the treat-to-target and of the tailored therapy for patients with autoinflammatory diseases contributes to consider glucocorticoids as a planned therapeutic approach. However, glucocorticoids must be prescribed for the shortest time necessary to obtain the remission or the minimal disease activity, reducing the cumulative dose. In fact, goals of the treatment in rheumatic diseases are symptom remission, prevention of complications, control of subclinical inflammation, and the achievement of an optimal quality of life. For these targets, treatment must be started soon, drugs need to be tailored to the patient, and the therapeutic strategy needs the planned partnership of clinicians, young patients, and families or caregivers [91].

However, the treat-to-target strategy can be guided by the lesson of the “neuroimmunoendocrinological” approach, with a strict endocrine, auxological, and metabolic follow-up of children with rheumatic diseases, especially when treated with glucocorticoids. Pediatric rheumatologists need to cooperate with pediatricians, pediatric neurologists, and pediatric endocrinologists. This team should evaluate the growth velocity, bone age, weight, body mass index, and BMD of their patients. However, a further health need is the supervision of the impact of chronic stress on metabolic and endocrine status and of the effects of glucocorticoid treatment on nictemeral rhythm and biological clock, which are milestones to preserve GH, FSH, and LH secretion.

Children and adolescents with rheumatic diseases need to be periodically checked with hormonal analysis of FSH, LH, IGF-1, ACTH, and cortisol. They need chemical analysis of fasting and post-prandial glycaemia, insulin, and C-peptide, as well as of vitamin D, alkaline phosphatase, calcium, phosphate, magnesium.

DEXA is a diagnostic evaluation to recognize patients at risk of fractures. This therapeutic strategy should follow the patient’s growth stages, to promptly prevent growth and pubertal delay before it clinically manifests. Furthermore, BMD needs nutritional support, with calcium and vitamin D supplementation, adequate for age and body weight.

## 6. Conclusions

This review highlights the key correlations among the CNS, endocrine, and immune systems in children and adolescents with rheumatic diseases. Glucocorticoids have a significant impact on this homeostatic balance, and this review proposed a holistic approach to steroid effects and the realization of a network of clinicians working to support these special patients.

Further studies in this direction can allow highlighting the role of glucocorticoid treatment in this scenario, as well as the possible optimal therapeutic choices to prevent side effects and long-term disabilities in children and adolescents with rheumatic diseases.

## Figures and Tables

**Figure 1 ijms-24-13192-f001:**
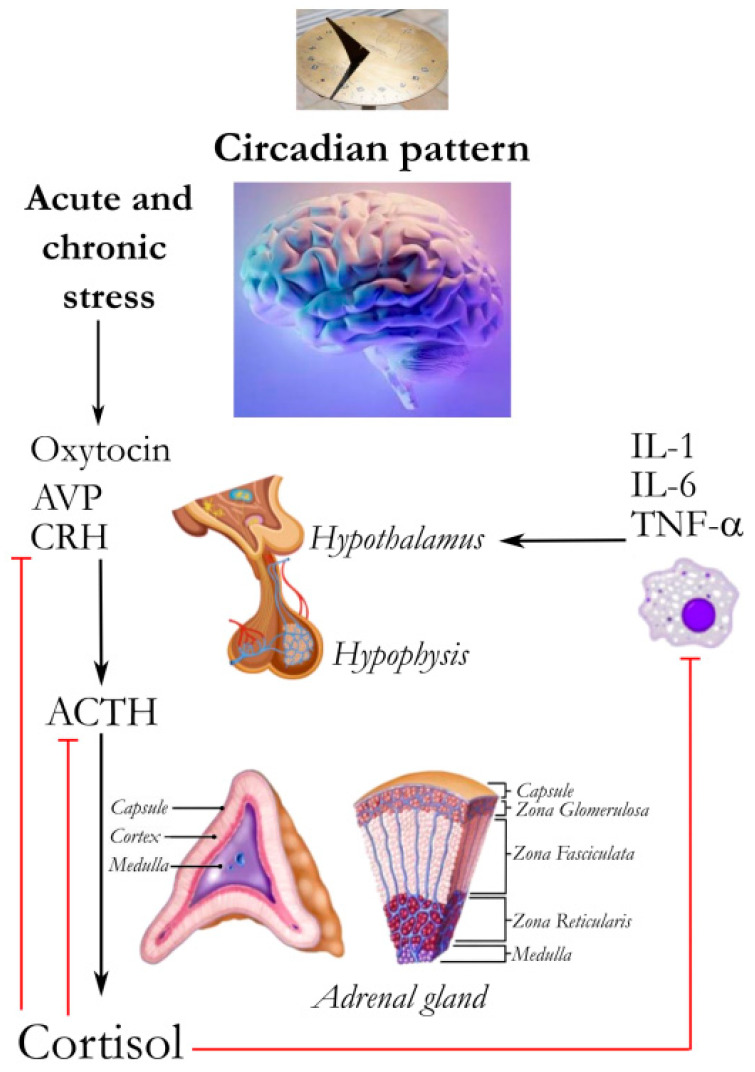
The network linking the nervous, immune, and endocrine systems. The circadian pattern and stress are influencers of central nervous system (CNS). Hormones and cytokines are important mediators of the hypothalamus–pituitary–adrenal (HPA) axis. Oxytocin may increase adrenocorticotropic hormone (ACTH) release, whereas atrial natriuretic peptide inhibits ACTH release via direct action on the pituitary. Cortisol and glucocorticoids act on their hypothalamic and pituitary receptors to suppress the release of arginine vasopressin (AVP), corticotropin-releasing hormone (CRH), and ACTH in response to these neuropeptides. Cytokines [interleukin (IL)-1, IL-6, and tumor necrosis factor (TNF)-α] stimulate the hypothalamus. Corticotropin-releasing hormone (CRH); adrenocorticotropic hormone (ACTH); arginine vasopressin (AVP); atrial natriuretic peptide (ANP); interleukin (IL)-1, IL-6; tumor necrosis factor (TNF)-α.

**Figure 2 ijms-24-13192-f002:**
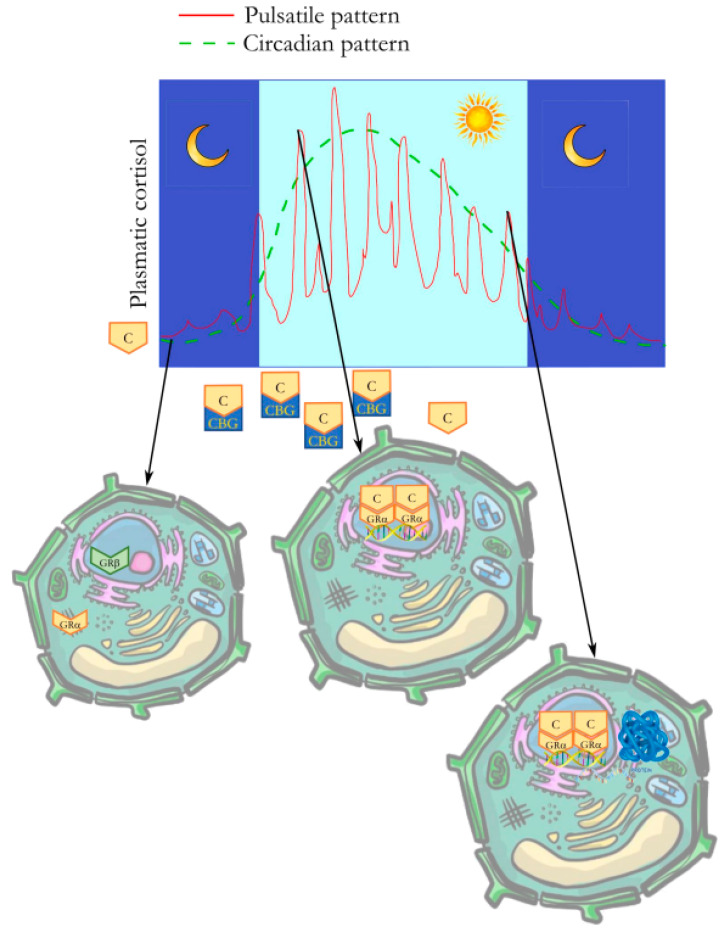
Glucocorticoid receptor signaling pathway. The glucocorticoid receptor (GR) signaling pathway shows a well-timed response to hormone levels, demonstrating that the regulation of gene targets by GR requires an ultradian pattern of hormone secretion. Cortisol (C) is transported in the blood predominantly bound to corticosteroid-binding globulin (CBG). CBG facilitates cortisol circulation and regulates its release to tissues. CBG-free cortisol passes passively across the plasma membrane. GRα, in the absence of the ligand, is in the cell’s cytoplasm and, after corticosteroid binding, is activated and translocated to the nucleus. It operates as a transcription factor. GR homodimers bind to DNA, allowing the receptors to regulate gene transcription. GRβ is expressed in the nucleus, does not bind corticosteroids, is inactive as a transcription factor, and inhibits the transcriptional activity of GRα.

**Table 1 ijms-24-13192-t001:** Corticosteroid comparison chart.

		Potency Relative to Hydrocortisone		Half-Life	
	Equivalent Glucocorticoid Dose (mg)	Anti-Inflammatory	Mineralocorticoid	Plasma (minutes)	Duration of Action (hours)
GLUCOCORTICOIDS					
Short acting					
Hydrocortisone	20	1	1	90	8–12
Cortisone acetate	25	0.8	0.8	30	8–12
Intermediate acting					
Prednisone	5	4	0.8	60	12–36
Prednisolone	5	4	0.8	200	12–36
Triamcinolone	4	5	0	300	12–36
Methylprednisolone	4	5	0.5	180	12–36
Long acting					
Dexamethasone	0.75	30	0	200	36–54
Betamethasone	0.6	30	0	300	36–54
MINERALCORTICOIDS					
Fludrocortisone	0	15	150	240	24–36
Aldosterone	0	0	>400	20	-

Prescribed steroid equivalents: prednisone 5 mg = cortisone 25 mg = dexamethasone 0.75 mg = hydrocortisone 20 mg (modified from [21]).

## Data Availability

The data of this paper are included in the references.

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
