# Peer review of "“Neuroimmunoendocrinology” in Children with Rheumatic Diseases: How Glucocorticoids Are the Orchestra Director"

_ijms, 2023, doi:10.3390/ijms241713192_

Round 1

Reviewer 1 Report

The manuscript titled: “NEUROIMMUNOENDOCRINOLOGY” IN CHILDREN WITH RHEUMATIC DISEASES: HOW GLUCOCORTICOIDS ARE THE ORCHESTRA DIRECTOR” represents insights into the impact of glucocorticoids on endocrine, immune, and neurologic targets. The authors describe the mechanism of homeostasis disruption by glucocorticosteroids (GCS)). The subject is worthy of representation; however, the review requires significant revision.

1.      The large paragraphs from lines 29 to 98, 163-174, 219-226, and 275-279, lack references. What is the source of the present information in this paragraph?

2.      Authors wrote: “The exposure to an excess of glucocorticoids inhibits GH secretion with growth delay in childhood and abnormalities in bone mineral density (BMD) and body composition in both children and adults. However, cortisol physiologically contributes to the maturation and function of somatotrophs, supporting the hypothesis that glucocorticoids act with a dose-dependent effect on the somatotropic axis.” (lines 330-333). Which dose of GCS can significantly decrease growth, maturation, BMD, 11β-HSD1 activity, and function of the HPA axis? Is the difference between the dose of 5 mg prednisone vs. 20 mg prednisone (or drug equivalent) on growth and body mass?

3.      What about inhaled steroids/ Do they influence NEUROIMMUNOENDOCRINOLOGY like systemic steroids?

4.      Many of the paragraphs are cited from one source, i.g, 603-622 – there is only one reference [69]. The authors mainly use already published reviews and do not discuss the role of GCS in the context of recent original studies. Thus, in my opinion, the review should be rewritten.

5.      What is the clinical meaning of described NEUROIMMUNOENDOCRINOLOGY – should clinicians prescribing GCS control some parameters in children (neurological – check the neurological function, immunological, endocrinological – e.g., hormonal)? Are the adverse effects dose-dependent?

6.      What is the novelty of your manuscript?

Author Response

The manuscript titled: “NEUROIMMUNOENDOCRINOLOGY” IN CHILDREN WITH RHEUMATIC DISEASES: HOW GLUCOCORTICOIDS ARE THE ORCHESTRA DIRECTOR” represents insights into the impact of glucocorticoids on endocrine, immune, and neurologic targets. The authors describe the mechanism of homeostasis disruption by glucocorticosteroids (GCS)). The subject is worthy of representation; however, the review requires significant revision.

  1. The large paragraphs from lines 29 to 98, 163-174, 219-226, and 275-279, lack references. What is the source of the present information in this paragraph?

We added references to these paragraphs, as suggested.

  1. Authors wrote: “The exposure to an excess of glucocorticoids inhibits GH secretion with growth delay in childhood and abnormalities in bone mineral density (BMD) and body composition in both children and adults. However, cortisol physiologically contributes to the maturation and function of somatotrophs, supporting the hypothesis that glucocorticoids act with a dose-dependent effect on the somatotropic axis.” (lines 330-333). Which dose of GCS can significantly decrease growth, maturation, BMD, 11β-HSD1 activity, and function of the HPA axis? Is the difference between the dose of 5 mg prednisone vs. 20 mg prednisone (or drug equivalent) on growth and body mass?

We developed these aspects in the chapter 4.1. Glucocorticoids and growth. 4.2 and Glucocorticoids and the bone.

  1. What about inhaled steroids/ Do they influence NEUROIMMUNOENDOCRINOLOGY like systemic steroids?

This topic needs a further review to study the influence of inhaled steroids on growth, puberty and BMD in children with asthma. However, in this paper we studied these aspects in children with rheumatic diseases.

  1. Many of the paragraphs are cited from one source, i.g, 603-622 – there is only one reference [69]. The authors mainly use already published reviews and do not discuss the role of GCS in the context of recent original studies. Thus, in my opinion, the review should be rewritten.

We added the references, as required. We discussed the role of GCS in the context of recent original studies, as required.

  1. What is the clinical meaning of described NEUROIMMUNOENDOCRINOLOGY – should clinicians prescribing GCS control some parameters in children (neurological – check the neurological function, immunological, endocrinological – e.g., hormonal)? Are the adverse effects dose-dependent?

We added and described these aspects in the “Discussion”.

  1. What is the novelty of your manuscript?

We followed the suggestion of the revisor and we added to the discussion the importance of the “neuroimmunoendocrinological” approach, with a strict endocrine, auxological and metabolic follow-up of children with rheumatic diseases, especially when treated with glucocorticoids. We highlighted the importance of a strict collaboration between paediatric rheumatologists, paediatricians, paediatric neurologists, and pediatric endocrinologists to evaluate growth velocity, bone age, weight, body mass index, BMD of their patients. This therapeutic strategy should follow patient's growth stages, to early prevent growth and pubertal delay before it becomes clinically manifest. Furthermore, BMD needs a nutritional support, with calcium and vitamin D supplementation, adequate to age and body weight.

Reviewer 2 Report

Dear Authors!

Thank you for the opportunity to read and reviewer the manuscript.

Despite the broad using of biologics and corticosteroid-free tendency in the treatment of immune-mediated disease the role of corticosteroids and their physiological and pathophysiological role still be interesting. This is a well written comprehensive review, covered different aspects of neuro-immune-endocrine regulation and the role of corticosteroids.

I recommend to add some more clinical information:

1. The table, comparing different corticosteroids, their efficacy, mechanism of action, doses and so on will be useful

2. Please add more information about bone avascular necrosis and the role of coricosteroids

3. Please add more information aboutassociation of corticosteroids and specific osteoarthritis, e.g. hip, shoulder.

4. Please add information about known cumulative cut-off doses of corticosteroids in low BMD, osteoporosis in immune-mediatd diseases

Minor English editing required.

Author Response

Comments and Suggestions for Authors

Dear Authors!

Thank you for the opportunity to read and reviewer the manuscript.

Despite the broad using of biologics and corticosteroid-free tendency in the treatment of immune-mediated disease the role of corticosteroids and their physiological and pathophysiological role still be interesting. This is a well written comprehensive review, covered different aspects of neuro-immune-endocrine regulation and the role of corticosteroids.

I recommend to add some more clinical information:

  1. The table, comparing different corticosteroids, their efficacy, mechanism of action, doses and so on will be useful

We added the table comparing the different corticosteroids, as suggested.

  1. Please add more information about bone avascular necrosis and the role of corticosteroids.

 This point was added in the paragraph 4.2. Glucocorticoids and the bone.

  1. Please add more information about association of corticosteroids and specific osteoarthritis, e.g. hip, shoulder.

We added more informations about this aspect, as required. Hip involvement must undergo to an accurate differential diagnosis, for the risk of  a neoplastic origin of the joint disease and OC lesions are more frequent in knees and hips, in patients who received multiple steroid injections and/or high doses of systemic glucocorticoids. However, shoulder arthritis is rare at the beginninig of JIA, but the incidence is higher during the follow-up. We added these data in the paragraph 4.2. Glucocorticoids and the bone.

  1. Please add information about known cumulative cut-off doses of corticosteroids in low BMD, osteoporosis in immune-mediated diseases

This point was added in the paragraph 4.2. Glucocorticoids and the bone.

Comments on the Quality of English Language

Minor English editing required.

We revised English, as suggested. Thank you.

Round 2

Reviewer 1 Report

Dear Authors,

Thank you for a revised version.

The the manuscript has been sufficiently improve.